# Epigenetic Modifier SETD8 as a Therapeutic Target for High-Grade Serous Ovarian Cancer

**DOI:** 10.3390/biom10121686

**Published:** 2020-12-16

**Authors:** Miku Wada, Asako Kukita, Kenbun Sone, Ryuji Hamamoto, Syuzo Kaneko, Masaaki Komatsu, Yu Takahashi, Futaba Inoue, Machiko Kojima, Harunori Honjoh, Ayumi Taguchi, Tomoko Kashiyama, Yuichiro Miyamoto, Michihiro Tanikawa, Tetsushi Tsuruga, Mayuyo Mori-Uchino, Osamu Wada-Hiraike, Yutaka Osuga, Tomoyuki Fujii

**Affiliations:** 1Department of Obstetrics and Gynecology, Faculty of Medicine, The University of Tokyo, 7-3-1 Hongo Bunkyo-ku, Tokyo 113-8655, Japan; wada-miku39@g.ecc.u-tokyo.ac.jp (M.W.); abibi1203a@gmail.com (A.K.); cherrypon11@gmail.com (Y.T.); futabainoue0315@gmail.com (F.I.); machiko.k0221@gmail.com (M.K.); harunorihonjoh@gmail.com (H.H.); ayumikidu246@gmail.com (A.T.); jemappelletomo@gmail.com (T.K.); ymiya-tky@umin.ac.jp (Y.M.); tanikawa-tky@umin.ac.jp (M.T.); tsurugatetsushi@gmail.com (T.T.); mayuyo1976@gmail.com (M.M.-U.); osamuwh-tky@umin.ac.jp (O.W.-H.); yutakaos-tky@umin.ac.jp (Y.O.); fujiit-tky@umin.ac.jp (T.F.); 2Division of Molecular Modification and Cancer Biology, National Cancer Center Research Institute, 5-1-1 Tsukiji, Chuo-ku, Tokyo 104-0045, Japan; ryuji.hamamoto@riken.jp (R.H.); sykaneko@ncc.go.jp (S.K.); maskomat@ncc.go.jp (M.K.); 3Cancer Translational Research Team, RIKEN Center for Advanced Intelligence Project, 1-4-1 Nihonbashi, Chuo-ku, Tokyo 103-0027, Japan

**Keywords:** epigenetic modifier, histone methyltransferase, SETD8, high-grade serous ovarian cancer, UNC0379, H4K20 monomethylation

## Abstract

The histone methyltransferase SETD8, which methylates the lysine 20 of histone H4 (H4K20), is reportedly involved in human carcinogenesis along with nonhistone proteins such as p53. However, its expression profiles and functions in the context of high-grade serous ovarian carcinoma (HGSOC) are still unknown. The purpose of this study was to investigate the role of SETD8 in HGSOC. We performed quantitative real-time PCR and immunohistochemistry to detect the expression of SETD8 in HGSOC samples and normal ovarian specimens. Then, we assessed the effect of the inhibition of SETD8 expression using small interfering RNA (siRNA) and a selective inhibitor (UNC0379) on cell proliferation and apoptosis in HGSOC cells. The expression of SETD8 was significantly upregulated in clinical ovarian cancer specimens compared to that in the corresponding normal ovary. In addition, suppression of SETD8 expression in HGSOC cells with either siRNA or UNC0379 resulted in reduced levels of H4K20 monomethylation, inhibition of cell proliferation, and induction of apoptosis. Furthermore, UNC0379 showed a long-term antitumor effect against HGSOC cells, as demonstrated by colony-formation assays. SETD8 thus constitutes a promising therapeutic target for HGSOC, warranting further functional studies.

## 1. Introduction

Patients with ovarian cancer have the poorest prognosis among all individuals with gynecological cancers. Over 2 × 10^6^ new cases of ovarian cancer are reported annually, along with ~1 × 10^5^ reported ovarian-cancer-related deaths [1].

Ovarian cancers are classified into different histological types: serous (high-grade and low-grade), clear cell, endometrioid, and mucinous carcinoma, among others. High-grade serous ovarian carcinoma (HGSOC) is one of the most deadly ovarian cancer types because patients with HGSOC are mainly diagnosed at an advanced stage of the disease [2]. Frequent gene alterations such as those in TP53 and BRCA1/2 are also observed in HGSOC cases. In particular, TP53 mutations are observed in almost all cases. Additionally, some patients with HGSOC harbor germline mutations in BRCA1/2, which are related to hereditary breast and ovarian cancer syndrome [3]. Notably, several molecular targeting drugs are currently approved for HGSOC, such as an anti-VEGF antibody and a poly (ADP-ribose) polymerase (PARP) inhibitor. Nevertheless, the mortality rate for HGSOC remains high. Therefore, it is necessary to develop new, more effective molecular targeting agents [3].

Molecular mechanisms of epigenetics include histone acetylation, phosphorylation, ubiquitination, sumoylation, and poly ADP-ribosylation, which regulate gene expression [4,5]. Notably, a large body of research suggests that the dysfunction of histone methylation in particular is involved in carcinogenesis and cancer progression [6]. Several histone methyltransferases and demethylases are reportedly frequently overexpressed in various types of cancers [7,8]. In our previous study, a histone methyltransferase, SUV39H2, was shown to induce chemo and radioresistance in lung cancer cells [9]. Additionally, among gynecological cancers, we reported that the histone methyltransferase SMYD2 is overexpressed in HGSOC cells and that the suppression of SMYD2 with either small interfering RNA (siRNA) or its selective inhibitor results in reduced growth and increased apoptosis [10]. Thus, these data indicate that the inhibition of histone methyltransferases and demethylases constitutes a promising novel strategy for cancer therapy.

The histone methyltransferase SETD8 methylates lysine 20 of histone 4 (H4K20). This epigenetic modification is involved in DNA replication, chromatin compaction, and mitotic condensation [11,12]. SETD8 also methylates nonhistone proteins, such as the tumor suppressor p53, PCNA, and Numb [13,14,15]. Previous studies showed that p53 methylation by SETD8 decreases its activity [13]. Overexpression of SETD8 has been observed in many types of tumors, such as bladder and lung cancers [14]. Moreover, high expression of SETD8 correlates with poor prognosis in patients with gastric cancer and esophageal cancer, suggesting that SETD8 may constitute a viable therapeutic target for various types of cancers [16,17,18,19]. For example, the use of a selective SETD8 inhibitor (UNC0379) resulted in a significant survival advantage in preclinical xenograft neuroblastoma models, suggesting SETD8 as a candidate therapeutic target in this cancer type [20,21]. However, to the best of our knowledge, there has been no report regarding the expression profiles and functional analysis of SETD8 in the context of HGSOC. Therefore, the goals of this study were to elucidate the involvement of SETD8 in HGSOC and the therapeutic potential of its inhibition. Toward this end, we first analyzed the expression of SETD8 in clinical HGSOC specimens and compared it against that in normal ovarian tissues. We then inhibited SETD8 in HGSOC cell lines to explore its effect on proliferation and apoptosis, which are cellular processes related to cancer development and progression. Our findings provide the basis for future investigation into the therapeutic potential of histone methyltransferases in HGSOC.

## 2. Materials and Methods

### 2.1. Fresh Frozen Clinical Samples

HGSOC tissue specimens (*n* = 34) and normal ovarian tissue specimens (*n* = 3) were obtained from patients receiving care at the University of Tokyo Hospital from 2010 to 2016. The Human Genome, Gene Analysis Research Ethics Committee of the University of Tokyo approved this study (approval number: 683-19) [10].

### 2.2. Cell Lines and Cell Culture Conditions

JHOS2, JHOS3, and JHOS4 cell lines (RIKEN Cell Bank, Ibaraki, Japan) were cultured in Dulbecco’s modified Eagle’s medium (DMEM)/F12 medium, with 10% heat-inactivated fetal bovine serum (FBS) and 1% penicillin/streptomycin. KURAMOCHI, OVSAHO, and OVKATE cell lines (JRCB, Osaka, Japan) were cultured in RPMI medium with 10% heat-inactivated FBS and 1% penicillin/streptomycin. The TYK-nu cell line (JCRB) was cultured in DMEM with the addition of 10% heat-inactivated FBS and 1% penicillin/streptomycin. The OVCAR3 cell line (ATCC, Manassas, VA, USA) was cultured in RPMI medium with the addition of 20% heat-inactivated FBS and 1% penicillin/streptomycin. We cultured all cells in an incubator at 37 °C in humidified air with 5% CO_2_.

### 2.3. Small Interfering RNA (siRNA) Transfection

siRNAs specific to SETD8 (siSETD8 #1: sense: 5′-GCAACUAGAGAGACAAAUC-3′ and antisense: 5′-GAUUUGUCUCUCUAGUUGC-3′; siSETD8 #2: sense: 5′-GAUUGAAAGUGGGAAGGAA-3′ and antisense: 5′-UUCCUUCCCACUUUCAAUC-3′) and MISSION siRNA Universal Negative Control (siNC) were purchased from Sigma-Aldrich (St. Louis, MO, USA). siRNAs (final concentration: 100 nM) were transfected with Lipofectamine-RNAi MAX Transfection Reagent (Invitrogen, Carlsbad, CA, USA).

### 2.4. RNA Extraction and Quantitative Real-Time Reverse Transcription-Polymerase Chain Reaction (qRT-PCR)

We performed qRT-PCR using a One-Step SYBR Prime Script RT-PCR Kit (TaKaRa Bio, Tokyo, Japan) in a Light Cycler instrument (Roche Diagnostics, Indianapolis, IN, USA). The sequences of primers (used at a final concentration of 10 pmol/µL) were as follows: SETD8 forward: 5′-AAGAAACGGGAGGCTCTGTACG-3′ and reverse: 5′-TCTAGTTGCATCCACGCAGTAG-3′; β-actin forward: 5′-CACACTGTGCCCATCTACGA-3′ and reverse: 5′-CTCCTTAATGTCACGCACGA-3′.

### 2.5. Cell Viability Assay

Cells were cultured in 96-well plates (2–3 × 10^3^ cells/well for siSETD8) and 24-well plates (2–6 × 10^3^ cells/well for UNC0379). After a 24 h incubation, cells were transfected with siSETD8 (final concentration: 100 nM) for 96–120 h or treated with a SETD8-selective inhibitor, UNC0379 (Sigma-Aldrich, #SML-1465), in full serum condition for 9 days. Then, we added 10% well volume of Cell Counting Kit-8 solution (Dojindo, Kumamoto, Japan) to each well and, after 2 h of incubation, measured the absorbance of the solution at 450 nm on a microplate reader (BioTek, Winooski, VT, USA).

### 2.6. Cell Cycle Analysis

After transfection with siSETD8 (100 nM) or treatment with UNC0379 (10 µM) for 96 h, cells were harvested with trypsin, washed twice with PBS, fixed with 70% ethanol, and incubated at 4 °C overnight. Next, we washed the cells twice with PBS, added an RNase A stock solution (final concentration: 0.5 mg/mL) to the cells, and then incubated them for 20 min at 37 °C. Cells were then treated with propidium iodide (PI, 50 mg/mL; Sigma-Aldrich) at 4 °C for 15 min in the dark and measured by fluorescence-activated cell sorting (FACS) with an Epics XL instrument (Beckman Coulter, Brea, CA, USA) using Cell Quest Pro software v3.1 (BD Bioscience, Franklin Lakes, NJ, USA).

### 2.7. Detection of Apoptosis

After transfection with siSETD8 (final concentration: 100 nM) or treatment with UNC0379 (10 μM) for 96 h, cells were harvested using trypsin. Then, we washed the pelleted cells twice with PBS, resuspended the pellet in 1× Binding Buffer, and stained the pelleted cells with fluorescein isothiocyanate (FITC), Annexin V, and PI (FITC Annexin V Apoptosis Detection kit II; BD Pharmingen) in the dark at room temperature for 15 min. After the addition of 1× Binding Buffer, we measured Annexin V-FITC/PI double-positive cells by flow cytometry.

### 2.8. Immunohistochemical Staining (IHC)

After the formalin-fixed, paraffin-embedded tissue sections were deparaffinized using xylene, microwave retrieval (600 watts, 5 min, six times) was performed using Target Retrieval Solution with a pH of 9 (Agilent, Santa Clara, CA, USA). Dako REAL Peroxidase-Blocking Solution was added and reacted with rabbit anti-SETD8 (#2996; Cell Signaling, MA, USA, diluted to 1:200) at 4 °C overnight. Tissue sections were stained using Dako REAL EnVision horseradish peroxidase conjugated rabbit/mouse secondary antibodies for 8 min and mounted with Dako REAL DAB + CHROMOGEN. Nuclei were stained by hematoxylin counterstaining. We defined the IHC score as the sum of the proportion score of positive cells (score 0: 0%, score 1: <1%, score 2: 1–10%, score 3: 10–33%, score 4: 33–67%, score 5: >67%) and the intensity score (IS) of staining of positive cells (score 0: background, score 1: weak staining, score 2: moderate staining, score 3: strong staining). We graded the IHC score as follows: 0–4, negative; 5–8, positive, according to a previous report [10].

### 2.9. Protein Extraction and Western Blotting

RIPA lysis buffer (Wako, Osaka, Japan, 188-02453) mixed with a protease inhibitor cocktail (Roche, 11836153001) was used for protein extraction. Proteins were loaded into the wells of a sodium dodecyl sulfate–polyacrylamide gel (BIO-RAD Mini-PROTEAN TGX Gels, CA, USA), subjected to electrophoresis, and transferred onto nitrocellulose membranes. Amersham ECL Select (GE Healthcare Life Sciences, Piscataway, NJ, USA) and ImageQuant LAS 4000 mini (GE Healthcare Life Sciences, Piscataway, NJ, USA) were used for detection. The antibodies used for Western blotting were as follows, with the dilution indicated in parenthesis: rabbit anti-SETD8 (#2996; Cell Signaling Technologies, 1:1000), rabbit polyclonal anti-histone H4 (mono methyl K20) (ab9051; Abcam, Cambridge, UK, 1:2000), and mouse anti-β-actin (A2228; Sigma-Aldrich, Darmstadt, Germany, 1:7000).

### 2.10. Colony Formation Assay

Cells were cultured in six-well plates (JHOS3: 2.0 × 10^3^ cells/well, OVCAR3: 4.0 × 10^3^ cells/well). After 24 h, cells were treated with UNC0379 (1 and 10 µM) for 10 days. Cells were fixed with 100% ethanol for 2 h and stained with 0.5% Giemsa (Wako) for 60 min, after which we counted the number of colonies (consisting of over 50 cells).

### 2.11. Statistical Analyses

Statistical significance was calculated by Student’s *t*-test and Pearson’s chi-square test using Excel and JMP Pro. v.12 (SAS, Cary, NC, USA). Western blot and densitometric analyses were quantified using the NIH ImageJ 1.52q software (NIH, Bethesda, Maryland, USA) (Figure 2C). The asterisks indicate different degrees of statistical significance as follows: ** *p* < 0.05 and * *p* < 0.01.

## 3. Results

### 3.1. Expression Profiling of Histone Methyltransferases Identifies SETD8 as Overexpressed in HGSOC Cell Lines and Tissues

To determine whether histone methyltransferases constitute an appropriate therapeutic target for HGSOC, we examined the expression levels of different histone methyltransferases in HGSOC specimens by quantitative real-time PCR (data not shown). The results showed that SETD8 was significantly overexpressed in HGSOC samples compared to the expression seen in normal ovarian tissue (Figure 1A). Immunohistochemical analysis of the tissue sections further showed that strong SETD8 staining could be observed in the nucleus of cancer cells. In contrast, weak or no staining was observed in normal ovaries (Figure 1B). These results indicated that SETD8 expression in HGSOC was elevated at both the protein and mRNA levels. Additionally, we examined the correlation of SETD8 mRNA expression with stage and prognosis, but found no clear correlation (Appendix A). Since the expression of SETD8 is higher in HGSOC tissue samples compared to that in normal tissues, it was concluded that the expression of SETD8 increased at the time of carcinogenesis.

### 3.2. SETD8 Is Involved in the Growth of HGSOC Cells through H4K20 Monomethylation

We initially determined the level of SETD8 expression in HGSOC cell lines by RT-PCR (Figure 2A). To explore whether overexpression of SETD8 is involved in the proliferation of HGSOC cells, we performed knockdown experiments in two HGSOC cell lines, JHOS3 and OVCAR3, using two independent siRNAs targeting SETD8 (siSETD8 #1 and #2) and a control siRNA (siNC).

The results of Western blotting analyses indicated that SETD8 expression at the protein level was significantly suppressed in SETD8-knockdown HGSOC cell lines (Figure 2C). Furthermore, upon SETD8 knockdown, we also observed a decrease in the monomethylation levels of histone H4 at lysine 20 (H4K20 me1) (Figure 2C). In addition, we performed a cell viability assay using the same knockdown strategy, from which we observed that HGSOC cell viability was reduced upon transfection of the SETD8 siRNAs (Figure 2B). Additionally, to evaluate the antitumor effect induced by SETD8 knockdown, we analyzed the SETD8 knockdown-induced effects on the cell cycle by flow cytometry. The increase in the proportion of sub-G1 phase cells in SETD8 siRNA-transfected HGSOC cells indicated an increase in apoptosis (Figure 3A). The percentage of apoptotic cells was also measured by an Annexin V-FITC/PI assay, the results of which further confirmed that SETD8 knockdown induced apoptosis in HGSOC (Figure 3B).

### 3.3. A SETD8-Selective Inhibitor Suppresses Cell Proliferation and Induces Apoptosis in HGSOC Cells

To examine the potential for therapeutic targeting of SETD8 in clinical practice in the near future, we treated eight HGSOC cell lines with a SETD8-selective inhibitor, UNC0379, and performed cell proliferation assays. The expression of SETD8 in HGSOC cell lines was examined as described in Section 3.2 (Figure 2A) to evaluate whether SETD8 expression might serve as a biomarker of the efficacy of UNC0379. We found that the growth suppression was dose-dependent, with a half-maximal inhibitory concentration (IC50) ranging from 0.39 to 3.20 µM (Figure 4A). This indicated that no apparent relationship existed between SETD8 expression and the effects of SETD8 inhibitors. Using this approach, we also evaluated whether the suppression of HGSOC cell growth was reflected by a reduction of H4K20me1 levels. Specifically, we evaluated JHOS3 and OVCAR3 cells in addition to HGSOC cell lines (TYK-nu) exhibiting low IC 50 (Figure 4B). In TYK-nu cell lines, there was a greater reduction in H4K20 methylation than in the other two cell lines (i.e., JHOS3 and OVCAR). In addition, colony formation assays showed that UNC0379 also attenuated the number of colonies formed by HGSOC cell lines, which is a measure of long-term proliferative capability (Figure 4C). Furthermore, the increase in the proportion of sub-G1 phase cells in UNC0379-treated HGSOC cells confirmed an increase in apoptosis (Figure 4D). Consistent with this, the Annexin V-FITC/PI assay suggested that SETD8 inhibition by UNC0379 suppressed cell proliferation through apoptosis (Figure 4E).

## 4. Discussion

Our data suggested that SETD8 was overexpressed in HGSOC specimens when compared with that in normal ovarian tissue. The results of in vitro experiments performed on different HGSOC cell lines further suggested that the overexpression of SETD8 is involved in promoting cell proliferation. Moreover, inhibiting SETD8 expression by genetic means or by using a small molecule inhibitor demonstrated the potential of SETD8 as a new therapeutic target in HGSOC.

SETD8 expression is enhanced in various carcinomas such as lung, renal, and gastric cancers [14,16,22]. Here, we propose a role for SETD8 in cancer development and/or progression in HGSOC by revealing for the first time its overexpression in this cancer type. Notably, although some individual ovarian cancer samples exhibited lower levels of SETD8 expression than that of the normal ovarian sample N1 based on qRT-PCR data, overall SETD8 expression was statistically significantly higher in ovarian cancer samples than in normal ovarian tissues. Additionally, IHC data also revealed that SETD8 was significantly overexpressed in HGSOC compared to normal ovarian samples.

There was no correlation between SETD8 expression and stage/prognosis. These data suggested that SETD8 might be involved in carcinogenesis.

In knockdown experiments with siRNA, we selected OVCAR3 cells as a SETD8 high expression cell line and JHOS3 cells, derived from a Japanese patient, as representative of low SETD8 expression. High-efficiency siRNA-mediated knockdown almost abolished SETD8 expression in both JHOS3 and OVCAR3 cells, irrespective of the original level of SETD8. Although SETD8 expression is unlikely to be a biomarker in SETD inhibitors, we first examined the relationship between SETD8 expression and IC50 values because we were unsure of the extent to which UNC0379 inhibits SETD8. The results of the knockdown and inhibitor experiments suggested that there may not be a correlation between the expression of SETD8 and its cytostatic effect. However, to prove this, additional studies, with an increased sample size or long-term administration, are needed. Additionally, the degree of SETD8 knockdown is largely correlated with a reduction in H4K20 methylation. In contrast, our previous report showed that the reduction in histone methyltransferase did not correlate with the changes in histone methylation in gynecological cancer [23]. Thus, we hypothesized that the correlation between siRNA-mediated histone methyltransferase suppression and the degree of histone methylation may depend on the type of cancer cell line and method of methyltransferase inhibition; however, further studies are needed to confirm this conjecture. Furthermore, SETD8 knockdown exhibited higher potency toward inhibiting cell proliferation and inducing apoptosis in OVCAR3 cells than in JHOS3 cells. The TYK-nu cells had exhibited the lowest IC50 of SETD8 selective inhibitors, and had the greatest reduction in H4K20 methylation among all cell lines studied (i.e., JHOS3 and OVCAR cells). These findings suggested that correlation might exist between H4K20 methylation and IC50 levels. However, further studies such as the experimental system for overexpressing SETD8 are needed to confirm whether SET8 overexpression contributes to increased H4K20 methylation and if it is involved in increasing the IC50 value.

Nevertheless, consistent with our findings, inhibition of SETD8 has also been reported to suppress the proliferation of glioma cells [20].

Based on our results, we propose that inhibition or knockdown of SETD8 suppresses the proliferation of HGSOC cells by inducing apoptosis. Histone methyltransferases methylate not only histone proteins but also nonhistone proteins and thereby regulate cellular functions. Thus, it was necessary to consider both pathways to explore the function of SETD8 in HGSOC. This induction of apoptosis could be explained by the following mechanism: SETD8 is known to suppress the function of p53 via p53 methylation [13]; hence, the suppression of SETD8 function could counteract p53 inhibition, resulting in the induction of apoptosis. However, we did not analyze p53 methylation in this study because the antimethylated p53 antibody designed by Shi et al. is not available for purchase. In addition, we hypothesized that different mechanisms triggered by p53 methylation are involved in the apoptosis observed in HGSOC, because ~90% of patients with HGSOC harbor TP53 loss-of-function mutations [3]. In fact, the HGSOC cell lines used in the present study, such as OVCAR3 and JHOS3, carry a p53 mutation that is therefore expected to have an inhibitory effect on cell proliferation through inhibition of SETD8 independent of the methylation of p53 [24]. Additionally, methylation of nonhistone proteins such as monomethylation of PCNA at lysine 248 and dimethylation of Numb at lysine 158 and 163 by SETD8 could be biomarkers for HGSOC [14,15]. Moreover, Veo et al. reported that H4K20 methylation by SETD8 controls downstream genes involved in tumor invasiveness, pluripotency, and cell proliferation [25]. These data suggest that H4K20 methylation can regulate multiple genes. Thus, we hypothesized that H4K20 methylation by SETD8 could be involved in cell proliferation and antiapoptosis in HGSOC cells. Consistent with our hypothesis, in our study, SETD8 suppression via knockdown or treatment with a selective inhibitor induced apoptosis. Although the inhibition of SETD8 has previously been shown to have a therapeutic effect in various cancer types [14,19,26,27], this is the first demonstration of this effect in HGSOC cells. In particular, in the colony formation assay, a potential long-term effect on cell proliferation was observed in HGSOC cells treated with a small molecule inhibitor of SETD8. Additionally, a polymorphism (rs 16917496) at the miR-502 binding site of the lysine methyltransferase 5A (SET8) and its correlation with colorectal cancer have been reported in a GWAS study on SETD8 and cancer [28].

Other SETD-domain protein methyltransferases, such as SETD3, have also been considered as therapeutic targets for cancer. For example, it has been reported that SETD3 negatively correlates with prognosis in breast cancers [29]. However, further clarification is required with regard to the role of this protein in HGSOC.

Our study has several limitations. First, although our data suggested that no relationship existed between SETD8 expression and the effects of SETD8 inhibitors, we did not assess biomarkers indicative of SETD8 inhibitor sensitivity in patients with HGSOC. Because inhibitors that target histone modifications have complex therapeutic effects, detailed analysis of chromatin structure, such as via ChIP-Seq and HiC-Seq, is required to search for biomarkers of histone methyltransferases inhibitors. Second, although blocking SETD8 functions resulted in both suppression of cell proliferation and reduction in H4K20 methylation, the specific downstream genes regulated by H4K20 methylation that are involved in cell proliferation and apoptosis in HGSOC remain to be identified. Toward this end, we plan to conduct a transcriptome analysis and chromatin immunoprecipitation sequencing experiments to investigate the gene pathway(s) involved in this mechanism. Finally, it is necessary to perform in vivo experiments such as those using patient-derived xenograft mice to test the genetic and pharmacological inhibition of SETD8 in order to further confirm the antitumor effects of SETD8 inhibitors in a more physiological context and improve the translational significance of the present study for HGSOC.

## 5. Conclusions

Our findings highlight that SETD8 is overexpressed in HGSOC, similar to observations made in other cancer types, suggesting that this methyltransferase might be involved in HGSOC progression. The use of selective SETD8 inhibitors such as UNC0379 may thus constitute a promising strategy to improve the prognosis of HGSOC (Appendix A).

## Figures and Tables

**Figure 1 biomolecules-10-01686-f001:**
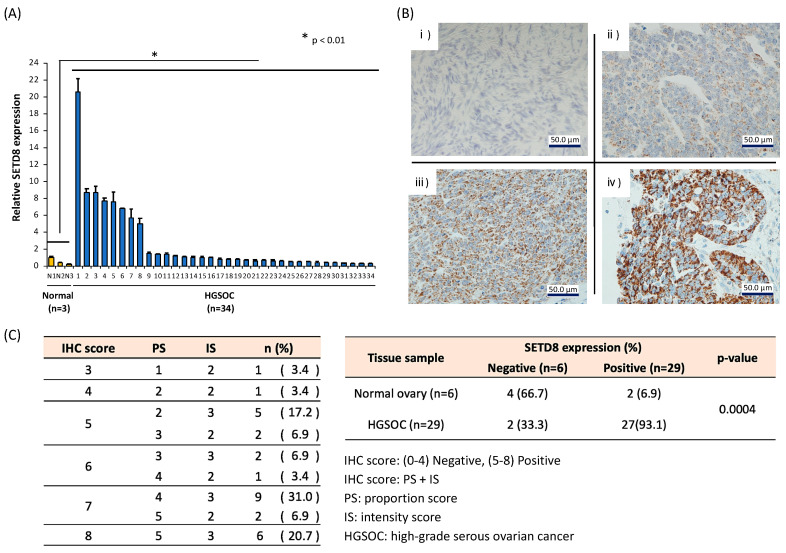
Elevated SETD8 expression in high-grade serous ovarian carcinoma (HGSOC). (**A**) mRNA levels of SETD8 were analyzed by quantitative polymerase chain reaction (qRT-PCR) in clinical high-grade serous ovarian carcinoma (HGSOC) specimens compared with those in normal ovarian tissues. The bars indicate the mean ± SD of three independent experiments (* *p* < 0.01). (**B**) Immunohistochemical (IHC) staining of SETD8 levels. Representative images of normal ovary and HGSOC tissues are shown. (**C**) Tables indicating the staining evaluation for all the samples. Intensity score (IS): (i) Normal ovary, score 0: background; (ii) HGSOC, score 1: weak staining; (iii) HGSOC, score 2: moderate staining; (iv) score 3: strong staining.

**Figure 2 biomolecules-10-01686-f002:**
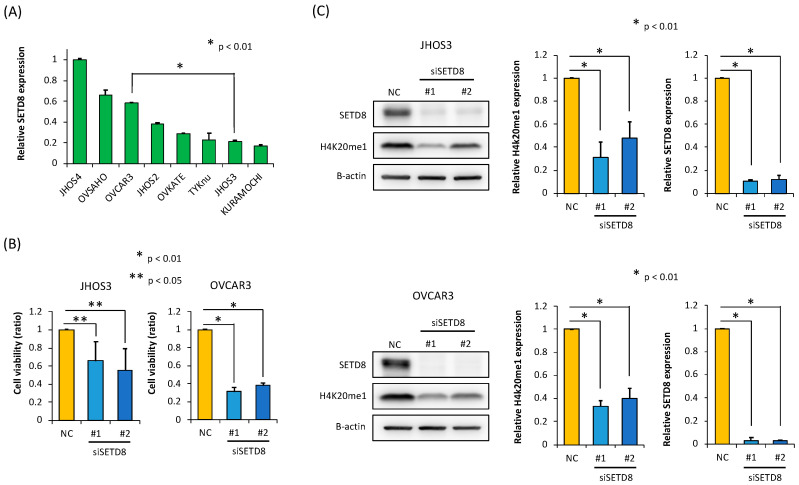
SETD8 knockdown reduces the cell viability of HGSOC cell lines. (**A**) The level of SETD8 expression in HGSOC cell lines by RT-PCR (* *p* < 0.01). (**B**) After the treatment of two HGSOC cell lines, JHOS3 and OVCAR3, with two different small interfering RNAs (siRNAs) targeting SETD8 (siSETD8#1 and siSETD#2) and control siRNA (siNC). Cell viability assays conducted 96–120 h after transfection with SETD8 siRNAs to evaluate growth suppression in JHOS3 and OVCAR3 cell lines (* *p* < 0.01). (**C**) After transfection with siSETD8#1, siSETD8#2, and siNC for 96 h, Western blot was carried out three times, and densitometric analysis of H4K20me1 and SETD8 protein levels normalized to β-actin protein levels in JHOS3 and TYK-nu cell lines were quantified using NIH ImageJ (* *p* < 0.01).

**Figure 3 biomolecules-10-01686-f003:**
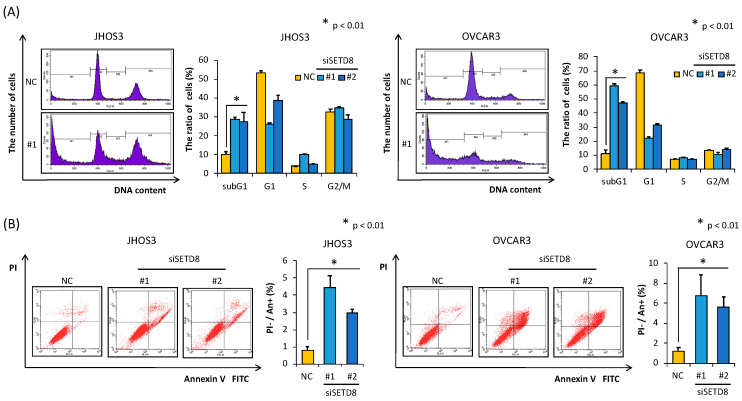
SETD8 knockdown induces apoptosis in HGSOC cells. (**A**) Cell cycle analysis showing the percentage of cells treated with siSETD8s or siNC in the sub-G1 phase. JHOS3 and OVCAR3 cells were treated with the indicated siRNAs for 96 h, and the cell cycle status was analyzed by flow cytometry and propidium iodide staining. The bars represent the mean ± SD of three independent experiments. (**B**) JHOS3 and OVCAR3 cells were treated with SETD8 siRNAs and siNC for 96 h, then the number of Annexin V positive cells was calculated by flow cytometry. The bars represent the mean ± SD of three independent experiments.

**Figure 4 biomolecules-10-01686-f004:**
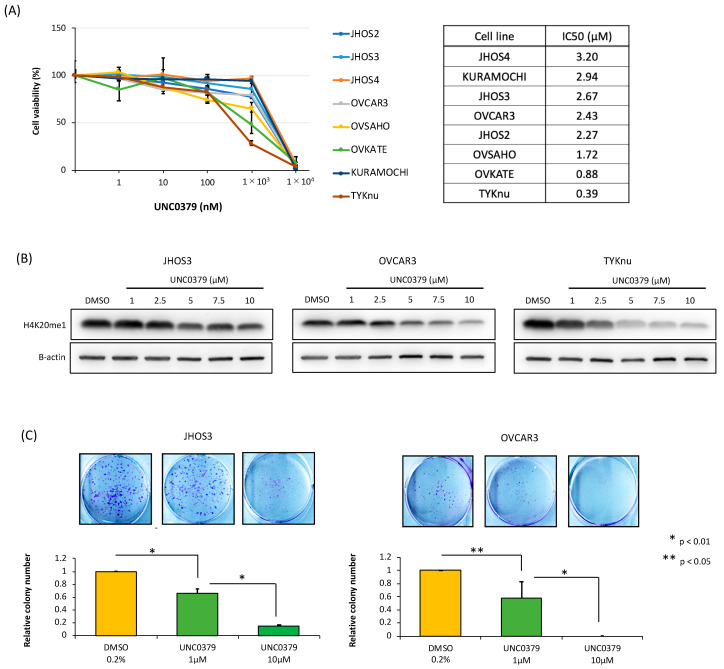
Effects of UNC0379, a SETD8-selective inhibitor, in HGSOC cell lines. (**A**) In vitro sensitivity of HGSOC cell lines to UNC0379. Following treatment with various concentrations of UNC0379 (1–10,000 nM) for 9 days in eight HGSOC cell lines, a cell viability assay was performed. The percentage of cell viability (%) was normalized in cells treated with 0.08% dimethyl sulfoxide (DMSO), used as control. (**B**) Representative Western blot images showing the dose-dependent reduction of H4K20 monomethylation in JHOS3, OVCAR3, and TYK-nu cells treated with UNC0379, whose concentration ranged from 0.1 to 10 µM, or 0.02% DMSO for 72 (TYK-nu) or 96 (JHOS3 and OVCAR3) hrs. Actin staining is shown as a loading control. (**C**) After the treatment of JHOS3 and OVCAR3 cells with UNC0379 at a concentration of 1 and 10 µM for 9 days, the number of colonies was manually counted and normalized to the number of colonies in cells treated with 0.2% DMSO. The bars represent the mean ± SD of three independent experiments (* *p* < 0.01, ** *p* < 0.05). (**D**) JHOS3 and OVCAR3 cells were treated with UNC0379 (10 µM) and DMSO for 96 h, and the cell cycle status was analyzed by flow cytometry and propidium iodide staining. The bars represent the mean ± SD of three independent experiments (* *p* < 0.01). (**E**) JHOS3 and OVCAR3 cells were treated with UNC0379 (10 µM) and DMSO for 96 h, and then the number of Annexin V positive cells was calculated using flow cytometry. The bars represent the mean ± SD of three independent experiments (* *p* < 0.01).

## Data Availability

All data generated and analyzed during this study are included in this published article.

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
