# Peer review of "Epigenetic Modifier SETD8 as a Therapeutic Target for High-Grade Serous Ovarian Cancer"

_biomolecules, 2020, doi:10.3390/biom10121686_

Round 1
Reviewer 1 Report
This manuscript described the expression pattern of SETD8 in high-grade serous ovarian cancer (HGSOC), and further inverstigated the function of SETD8 bu using siRNA and chemical inhibitor UNC0379. In general, the present study reported an interesting finding with potential benefit for benifit for HGSOC. A few questions should be addressed before publication:
1, Since the expression level of SETD8 was quite variable in different HGSOC patients, was there any correlation between the expression level of SETD8 and the grade/survival/ of patients? For example, any differences between those patient with high expression level of SETD8 and those with low level?
2, Similar issue for the cell lines. The authors chose two cell lines with different levels of SETD8, which was good, but it was difficult to understand why knockdown of SETD8 in JHOS3 with lower endogenious SETD8 level also resulted in simialrly significant effect to OVCAR3.
3, As mentioned by the authors, no correlation between in expression (Fig.2A) and the IC50 (Fig.4A). The authors should discuss the possibility, if no experimental data provided.
4, The authors determined the H4K20me1 level after the treatment of UNC0379, suggesting the epigenetic regulation might contribute to the phenotype. Experiment of overexpressing wild-type SETD8 or enzymatically mutated version would provide direct evidence. In addition, it would be very interesting that what would happen in the cell line JHOS3 with lower SETD8 if overexpressing exogenious SETD8?
Author Response
We appreciate your critical comments and useful suggestions. Please note that the changes made do not influence the content, conclusions, or framework of the paper. The main changes have been highlighted in yellow in the revised manuscript.
Reviewer 1
This manuscript described the expression pattern of SETD8 in high-grade serous ovarian cancer (HGSOC), and further inverstigated the function of SETD8 bu using siRNA and chemical inhibitor UNC0379. In general, the present study reported an interesting finding with potential benefit for benifit for HGSOC. A few questions should be addressed before publication:
Comment1
Since the expression level of SETD8 was quite variable in different HGSOC patients, was there any correlation between the expression level of SETD8 and the grade/survival/ of patients? For example, any differences between those patient with high expression level of SETD8 and those with low level?
Response 1: As suggested by you, we examined the correlation between SETD8 expression and stage and prognosis, but found no clear correlation. ​Since the expression of SETD8 is higher in HGSOC tissue samples than that in normal tissues, it was considered that expression of SETD8 increased at the time of carcinogenesis. The abovementioned consideration has been added to the Result and Discussion section (Lines 173–176, 269–270).
Comment2
Similar issue for the cell lines. The authors chose two cell lines with different levels of SETD8, which was good, but it was difficult to understand why knockdown of SETD8 in JHOS3 with lower endogenious SETD8 level also resulted in simialrly significant effect to OVCAR3.
Response 2: High efficiency siRNA-mediated knockdown abolished SETD8 expression almost completely in both JHOS3 and OVCAR3 cells, irrespective of the original level of SETD8. In addition, there may not be a correlation between the expression of SETD8 and its cytostatic effect with knockdown of SETD8. The statement mentioned above has been added to the Discussion section (Lines 278–281).
Comment3
As mentioned by the authors, no correlation between in expression (Fig.2A) and the IC50 (Fig.4A). The authors should discuss the possibility, if no experimental data provided.
Response 3: Our data showed no clear relationship between SETD8 expression and IC50 levels, which may be related to H4K20 methylation. ​However, to prove this, additional studies such as with an increased sample size or long-term administration are needed. The statement mentioned above has been added to the Discussion section (Lines 278–281).
Comment4
The authors determined the H4K20me1 level after the treatment of UNC0379, suggesting the epigenetic regulation might contribute to the phenotype. Experiment of overexpressing wild-type SETD8 or enzymatically mutated version would provide direct evidence. In addition, it would be very interesting that what would happen in the cell line JHOS3 with lower SETD8 if overexpressing exogenious SETD8?
Response 4: As noted by you, the experimental system for overexpressing SETD8 is very interesting. ​Our data showed that SETD8 expression was not associated with UNC0379 IC50 values. ​Thus, overexpression of SETD8 may not affect this phenotype. ​However, if SET8 overexpression contributes to increased H4K20 methylation, it may also affect the phenotypes. This information has been added to the Discussion section (Lines 290–294).
Reviewer 2 Report
Wada et al. study is aimed at demonstrating the role of SETD8 and its genetic and pharmacological inhibition by using the UNC0379 compound in the High-grade Serous Ovarian Cancer (HGSOC) tumorigenesis. The authors demonstrated that SETD8 is overexpressed in tissues derived from patients affected by HGSOC compared with normal ovarian tissue. Inhibition of SETD8 impairs HGSOC cell line viability and induces apoptosis. The authors propose the targeting of SETD8 as a potential novel therapeutic approach for HGSOC patients.
The following concerns need to be addressed:
Major concerns:
-The title of the manuscript “Overexpression of the Epigenetic Modifier SETD8 in the Proliferation of Cancer Cells: the SETD8 Inhibitor, UNC0379 as a Potent Therapeutic Drug for High-grade Serous Ovarian Cancer” is not completely clear. The authors should better characterize the main message of the manuscript. I would suggest the following title: “Epigenetic Modifier SETD8 as a Therapeutic target for High-grade Serous Ovarian Cancer”.
-The sentence at line 221 “…. to evaluate whether SETD8 expression might serve as a biomarker of the UNC0379 activity” is not completely correct. The authors should better explain why they used SETD8 expression as a biomarker of UNC0379 efficacy. Given that UNC0379 is a substrate competitive inhibitor, it should not have impact on SETD8 protein expression levels. UNC0379 activity should be evaluated only based on its epigenetic marks as H4K20me1 and p53K382me1, and the other non-histone marks. However, it has been reported a down-regulation of SETD8 protein expression levels upon the treatment with UNC0379 also in other cancers, but the mechanisms are still unknown.
- In the Discussion section the authors should clarify the sentences starting from line 273 to 280. To this reviewer is not completely clear which results show what the authors are claiming here.
-The authors should have evaluated other SETD8 non-histone marks considering that the effect of SETD8 pharmacological or genetic inhibition may not only depend on H4K20me1. Although, the authors mention the SETD8 additional targets in the Discussion section, focusing in particular on the potential p53K382me1 role in apoptosis induction, they should also comment the potential effect of the other non-histone marks (PCNA, NUMB).
-In vivo experiments testing the genetic and pharmacological inhibition of SETD8 in HGSOC xenograft models could be added to improve the translational significance of the study.
Minor concerns:
-The authors should improve the image quality of Fig 1B (IHC images) and insert properly scale bars, which are too small.
-A scheme illustrating the novel findings of SETD8 role in HGSOC and its potential mechanism of action (by a pro-apoptotic response induction) could be added.
Author Response
We appreciate your critical comments and useful suggestions. Please note that the changes made do not influence the content, conclusions, or framework of the paper. The main changes have been highlighted in yellow in the revised manuscript.
Wada et al. study is aimed at demonstrating the role of SETD8 and its genetic and pharmacological inhibition by using the UNC0379 compound in the High-grade Serous Ovarian Cancer (HGSOC) tumorigenesis. The authors demonstrated that SETD8 is overexpressed in tissues derived from patients affected by HGSOC compared with normal ovarian tissue. Inhibition of SETD8 impairs HGSOC cell line viability and induces apoptosis. The authors propose the targeting of SETD8 as a potential novel therapeutic approach for HGSOC patients.
The following concerns need to be addressed:
Major concerns:
Comment1-The title of the manuscript “Overexpression of the Epigenetic Modifier SETD8 in the Proliferation of Cancer Cells: the SETD8 Inhibitor, UNC0379 as a Potent Therapeutic Drug for High-grade Serous Ovarian Cancer” is not completely clear. The authors should better characterize the main message of the manuscript. I would suggest the following title: “Epigenetic Modifier SETD8 as a Therapeutic target for High-grade Serous Ovarian Cancer”.
Response 1: We have revised the title accordingly.
Comment 2-The sentence at line 221 “…. to evaluate whether SETD8 expression might serve as a biomarker of the UNC0379 activity” is not completely correct. The authors should better explain why they used SETD8 expression as a biomarker of UNC0379 efficacy. Given that UNC0379 is a substrate competitive inhibitor, it should not have impact on SETD8 protein expression levels. UNC0379 activity should be evaluated only based on its epigenetic marks as H4K20me1 and p53K382me1, and the other non-histone marks. However, it has been reported a down-regulation of SETD8 protein expression levels upon the treatment with UNC0379 also in other cancers, but the mechanisms are still unknown.
Response 2: Initially we were unsure of the extent to which UNC0379 inhibits SETD8, therefore, we first examined the relationship between SETD8 expression and IC50 values. As you mentioned, our results suggest that SETD8 expression is unlikely to be a biomarker and that H4K20 methylation may be a biomarker instead. ​Methylation of non-histone proteins such as PCNA and NUMB may also become a biomarker. In contrast, P53 methylation may not be informative as p53 is mutated in HGSCO. This information has been added to the Discussion section (Lines 276–279 and 310–311).
Comment 3- In the Discussion section the authors should clarify the sentences starting from line 273 to 280. To this reviewer is not completely clear which results show what the authors are claiming here.
Response 3: As knockdown of SETD8 reduces H4K 20 methylation, we have revised the test as follows: "Additionally, the degree of SETD8 knockdown is largely correlated with a reduction in H4K20 methylation."​ This information has been added to the Discussion (Lines 281–283).
Comment 4-The authors should have evaluated other SETD8 non-histone marks considering that the effect of SETD8 pharmacological or genetic inhibition may not only depend on H4K20me1. Although, the authors mention the SETD8 additional targets in the Discussion section, focusing in particular on the potential p53K382me1 role in apoptosis induction, they should also comment the potential effect of the other non-histone marks (PCNA, NUMB).
Response 4 As you noted, methylation of non-histone proteins other than p53 may have an effect. ​We have added some references to the Discussion section (Lines 310–311).
Comment 5-In vivo experiments testing the genetic and pharmacological inhibition of SETD8 in HGSOC xenograft models could be added to improve the translational significance of the study.
Response 5: ​As noted, in vivo experiments are necessary to improve the significance of this study. We plan to conduct these in vivo experiments in the future. ​This was added as a limitation in this paper (Lines 338–341).
Minor concerns:
-The authors should improve the image quality of Fig 1B (IHC images) and insert properly scale bars, which are too small.
Response: We have revised the figures accordingly.
-A scheme illustrating the novel findings of SETD8 role in HGSOC and its potential mechanism of action (by a pro-apoptotic response induction) could be added.
Response: We have added a scheme illustrating the novel findings of SETD8 role in HGSOC and its potential mechanism of action.
Round 2
Reviewer 1 Report
This manuscript reported an interesting finding that UNC0379 might be a potential drug forcertain HGSOC subpopulation, but also raised a number of remaining issues regarding the relevance to SETD8/H4K20me and the underlying mechanism.
This manuscript is a resubmission of an earlier submission. The following is a list of the peer review reports and author responses from that submission.
Round 1
Reviewer 1 Report
Summary:
SETD8 was examined in HGSOC cells and, similar to observation in other cancer types, was elevated in HGSOC compared to normal tissue. Some tumors had a very large increase in SETD8 compared to normal tissue, while many of the tumors examined had a much less robust increase compared to normal tissues. This suggests that there is a great variation of SETD8 expression in HGSOC. The manuscript also shows that the viability of 2 model cell lines for HGSOC have decreased viability upon siRNA knockdown of SETD8 and 8 model cell lines had decreased viability upon small molecule competitive inhibition of SETD8. Cell cycle analysis and apoptosis was also quantified in these studies. However, a more mechanistic analysis (described below) is necessary to make this manuscript competitive for publication.
Major Issues:
- The data in Figure 1A is contradictory to the conclusion “all 35 HGSOC tissues had upregulation of SETD8”. The data in figure 1A shows that the transcript level of SETD8 of N1 seems higher than HGSOC tissue numbers 18 to 35. If this conclusion was based on the statistical difference calculated, the method of statistical analysis should be clearly stated in the text. Which method was used for this specific set of data? It is unclear if authors used different specimens for the analysis shown in Figure 1A and Figures 1C. If the same tissues were analyzed for A and C, then the authors must explain why there are different sample numbers for each type of analysis. Measure methylated histone on the same samples that immunohistochemistry was performed for SETD3 staining.
- Figure 4 shows data using 8 HCSOC cell lines. Only 2 (OVCAR3 and JHOS3) were selected for initial analysis of SETD8 expression/knockdown followed by cell viability assays in Figure 2. The authors must justify why these 2 were chosen over the other 6 cell lines. To further investigate the impact and importance of SETD8, I would suggest that the authors should include one additional human ovarian cancer cells with low or normal expression of SETD8 and for the experiment. Furthermore, in Figure 2a, the reduction of SETD8 expression by siRNA does not is not correlate with changes in monomethylation of the target histone. SETD8 is still expressed in JHOS3 cells after knockdown while there is an undetectable level of histone monomethylation. This needs to be explained. Data showing relative protein expression for SETD8 for all 8 cell lines should be included in Figure 2. If there is a cell line that has relatively low SETD8 expression, it should be included in all experiments shown in Figure 2. OVKATE and TYKnu cell lines had a very different response to SETD8 inhibition than the others. Monomethylation inhibition should be analyzed OVKATE or TYKu cell that had a very different response to SETD8 inhibition. This would add depth to the study regarding the correlation between inhibition of SETD8 and HGOCG viability. Why were JHOS3 the only cells that were measured decreased histone methylation after inhibitor treatment. This must be explained.
- SETD8 methylates p53 and this was mentioned in the introduction. p53 methylation should be measured and compared with SETD8 expression/inhibition to tie this into the mechanism of apoptosis regulation. Is p53 methylated in the cell lines tested? Is methylation reduced after SETD8 knockdown and/or inhibition? These mechanistic studies need to be performed for this current manuscript.
Minor issues:
Line 84: the percentage of FBS is missing.
Line 108-109: please specify that If the treatment was performed in serum-free, serum-starved, or full serum condition with UNC0379.
Line 150 “cells were added to”Figure 2b should not describe results as growth suppression, this should be consistent with the results sections and figure label “cell viability”
Line 182: the font of HGSOC cell lines is incorrect.
Line 195 and 196, grammar issues
Line 213: the unit of IC50 is incorrect, and it should be µM.
Figure 3 A and B: Missing the flow cytometry figure of JHOS3 cell. Explain why these data were not included?
Figure 3a resolution is inadequate to read the labels within the plot. I assume these are the gates used for the counts shown in the 3b.
Line 222 “cance” is incorrect
What statistics were performed for the IC50 plots. How many replicates were performed? This needs to be clear in the figure legend.
Author Response
Reviewer 1
Major Issues:
Comment1
The data in Figure 1A is contradictory to the conclusion “all 35 HGSOC tissues had upregulation of SETD8”. The data in figure 1A shows that the transcript level of SETD8 of N1 seems higher than HGSOC tissue numbers 18 to 35. If this conclusion was based on the statistical difference calculated, the method of statistical analysis should be clearly stated in the text. Which method was used for this specific set of data? It is unclear if authors used different specimens for the analysis shown in Figure 1A and Figures 1C. If the same tissues were analyzed for A and C, then the authors must explain why there are different sample numbers for each type of analysis. Measure methylated histone on the same samples that immunohistochemistry was performed for SETD3 staining.
Reply 1
We appreciate your critical comments and useful suggestions. Although some ovarian cancer samples expressed lower levels of SETD8 than did normal ovarian sample N1, statistical tests of SETD8 expression showed a significantly higher level of SETD8 expression in ovarian cancer samples than in normal ovarian tissue. We changed the wording based on your suggestion. The above analysis method is described in the method section. Of the 29 cases of IHC HGSOC, 24 cases were the same as those used in qPCR. The 5 cases used in the other IHC were not used for RT-PCR because the PCR did not work well. Associations between SETD3 and cancer have been reported recently. We also will plan to analyze the relationship between SETD3 and ovarian cancer in the future.
Comment2
Figure 4 shows data using 8 HCSOC cell lines. Only 2 (OVCAR3 and JHOS3) were selected for initial analysis of SETD8 expression/knockdown followed by cell viability assays in Figure 2. The authors must justify why these 2 were chosen over the other 6 cell lines. To further investigate the impact and importance of SETD8, I would suggest that the authors should include one additional human ovarian cancer cells with low or normal expression of SETD8 and for the experiment. Furthermore, in Figure 2a, the reduction of SETD8 expression by siRNA does not is not correlate with changes in monomethylation of the target histone. SETD8 is still expressed in JHOS3 cells after knockdown while there is an undetectable level of histone monomethylation. This needs to be explained. Data showing relative protein expression for SETD8 for all 8 cell lines should be included in Figure 2. If there is a cell line that has relatively low SETD8 expression, it should be included in all experiments shown in Figure 2. OVKATE and TYKnu cell lines had a very different response to SETD8 inhibition than the others. Monomethylation inhibition should be analyzed OVKATE or TYKu cell that had a very different response to SETD8 inhibition. This would add depth to the study regarding the correlation between inhibition of SETD8 and HGOCG viability. Why were JHOS3 the only cells that were measured decreased histone methylation after inhibitor treatment. This must be explained.
Reply2
We appreciate your critical comments and useful suggestions.
In Figure 2, we chose two ovarian cancer cell lines because of their cost and ease of handling.
We have also provided the results of SETD8 expression analysis (RT-PCR) in HGSOC cell lines as a Supplemental Figure. Th data suggest that there is no relationship between SETD8 expression and the effects of SETD8 inhibitors. Our previous data also showed no relationship between histone methylases and the effects of their inhibitors. Thus, new biomarkers need to be developed.
We hypothesized that the reason for the lack of correlation between suppression of SETD8 by siRNA and the expression of mono-methylation depends on the type of cell line, but further studies are needed to prove this. In place of SETD8 protein expression in cell lines, mRNA expression is shown as additional data. Since the expression of SETD8 does not correlate with the effect of the inhibitor as described above, it is assumed that the degree of methylation does not correlate with the effect of the inhibitor as well, but further investigation is required.
As for the reason why only JHOS3 was examined for methylation, the cell line used the most for analysis in this study and easy to handle was chosen as a representative example.
Comment3
SETD8 methylates p53 and this was mentioned in the introduction. p53 methylation should be measured and compared with SETD8 expression/inhibition to tie this into the mechanism of apoptosis regulation. Is p53 methylated in the cell lines tested? Is methylation reduced after SETD8 knockdown and/or inhibition? These mechanistic studies need to be performed for this current manuscript.
Reply3
We appreciate your critical comments and useful suggestions. Although a previous study reported P53 methylation using an antibody, which was designed by Shi, X et al. the antibody is not available for purchase. Therefore, we did not analyze p 53 methylation in this study.
Minor issues
Line 84: the percentage of FBS is missing.
Line 108-109: please specify that If the treatment was performed in serum-free, serum-starved, or full serum condition with UNC0379.
Line 150 “cells were added to”Figure 2b should not describe results as growth suppression, this should be consistent with the results sections and figure label “cell viability”
Line 182: the font of HGSOC cell lines is incorrect.
Line 195 and 196, grammar issues
Line 213: the unit of IC50 is incorrect, and it should be µM.
Figure 3 A and B: Missing the flow cytometry figure of JHOS3 cell. Explain why these data were not included?
Figure 3a resolution is inadequate to read the labels within the plot. I assume these are the gates used for the counts shown in the 3b.
Line 222 “cance” is incorrect
We appreciate your critical comments and useful suggestions.
We have revised the draft and figure as suggested to address these issues.
Reviewer 2 Report
Generally speaking, this paper is well written. However, there are several points that should be noted:
- In page 4, the regular expression of Figure 1 shuld be revised, such as *p<0.05 and **p<0.01.
- The color of Figure 3 (In page 6) should be corrected (As well as Figure 1).
- Figure 4E is no mentioned in the maintext.
- The format of References are not complete and are incosistent. Such as Ref3. (Page is missing), Ref10. (The format is inconsistent) and Ref 22.
Author Response
There are two main possible tumor mechanisms of UNC0379, which mediates histone methylation to induce apoptosis-related genes and inhibit p 53 methylation to activate p 53. Further examination is necessary to prove this. Polymorphism (rs 16917496) at the miR -502 binding site of the lysine methyltransferase 5A (SET8) and its correlation with colorectal cancer has been reported by a GWAS study on SETD8 and cancer.We will this information and the related discussion.
Reviewer 3 Report
this is a straightforward careful study. my only suggestions would be to add a description of how UNCO379 works or is thought to work. I would also like to know if there are any GWAS studies linking the SETD8 gene and cancer. This information could be added to the intro or discussion.
Author Response
We appreciate your critical comments and useful suggestions.
We have revised the draft and figure as suggested to address these issues.
Round 2
Reviewer 1 Report
The concerns of this reviewer was not addressed in the revised manuscript.
Author Response
Prof. Bei Zhang
Prof. Enric Sayas
Managing Editors
Biomolecules
Dear Editors
We wish to submit our revised manuscript titled “Overexpression of the epigenetic modifier, SETD8, in the proliferation of cancer cells: a potential therapeutic target in high-grade serous ovarian cancer” for publication in Biomolecules.
We have highlighted the main changes in the text in green.
Reviewer 1
Comment (Round2)
Major Issues: The concerns of this reviewer was not addressed in the revised manuscript.
Reply
We are very grateful to the reviewer for carefully reviewing our manuscript and for their valuable comments that have greatly helped us to improve the manuscript. We have addressed
the concerns of revewer1 in the revised manuscript. We addressed the revised sentences as bellows. We have highlighted the main changes in the text in green.
Comment1(Round1)
The data in Figure 1A is contradictory to the conclusion “all 35 HGSOC tissues had upregulation of SETD8”. The data in figure 1A shows that the transcript level of SETD8 of N1 seems higher than HGSOC tissue numbers 18 to 35. If this conclusion was based on the statistical difference calculated, the method of statistical analysis should be clearly stated in the text. Which method was used for this specific set of data? It is unclear if authors used different specimens for the analysis shown in Figure 1A and Figures 1C. If the same tissues were analyzed for A and C, then the authors must explain why there are different sample numbers for each type of analysis. Measure methylated histone on the same samples that immunohistochemistry was performed for SETD3 staining.
Reply 1
We appreciate your critical comments and useful suggestions. Although some ovarian cancer samples expressed lower levels of SETD8 than did normal ovarian sample N1, statistical tests of SETD8 expression showed a significantly higher level of SETD8 expression in ovarian cancer samples than in normal ovarian tissue. The above consideration was added to the discussion (Lines 248-251). We changed the wording based on your suggestion. The above analysis method is described in the method section. Of the 29 cases of IHC HGSOC, 24 cases were the same as those used in qPCR. The 5 cases used in the other IHC were not used for RT-PCR because the PCR did not work well. Associations between SETD3 and cancer have been reported recently. We also will plan to analyze the relationship between SETD3 and ovarian cancer in the future. The sentences about SETD3 was added to the discussion (Lines 278-279)
Comment2 (Round1)
Figure 4 shows data using 8 HCSOC cell lines. Only 2 (OVCAR3 and JHOS3) were selected for initial analysis of SETD8 expression/knockdown followed by cell viability assays in Figure 2. The authors must justify why these 2 were chosen over the other 6 cell lines. To further investigate the impact and importance of SETD8, I would suggest that the authors should include one additional human ovarian cancer cells with low or normal expression of SETD8 and for the experiment. Furthermore, in Figure 2a, the reduction of SETD8 expression by siRNA does not is not correlate with changes in monomethylation of the target histone. SETD8 is still expressed in JHOS3 cells after knockdown while there is an undetectable level of histone monomethylation. This needs to be explained. Data showing relative protein expression for SETD8 for all 8 cell lines should be included in Figure 2. If there is a cell line that has relatively low SETD8 expression, it should be included in all experiments shown in Figure 2. OVKATE and TYKnu cell lines had a very different response to SETD8 inhibition than the others. Monomethylation inhibition should be analyzed OVKATE or TYKu cell that had a very different response to SETD8 inhibition. This would add depth to the study regarding the correlation between inhibition of SETD8 and HGOCG viability. Why were JHOS3 the only cells that were measured decreased histone methylation after inhibitor treatment. This must be explained.
Reply2
We appreciate your critical comments and useful suggestions.
In Figure 2, we chose two ovarian cancer cell lines because of their cost and ease of handling.
We have also provided the results of SETD8 expression analysis (RT-PCR) in HGSOC cell lines as a Supplemental Figure. Th data suggest that there is no relationship between SETD8 expression and the effects of SETD8 inhibitors. Our previous data also showed no relationship between histone methylases and the effects of their inhibitors. Thus, new biomarkers need to be developed. The above consideration was added to the discussion (Lines 278-279).
We hypothesized that the reason for the lack of correlation between suppression of SETD8 by siRNA and the expression of mono-methylation depends on the type of cell line, but further studies are needed to prove this. In place of SETD8 protein expression in cell lines, mRNA expression is shown as additional data. Since the expression of SETD8 does not correlate with the effect of the inhibitor as described above, it is assumed that the degree of methylation does not correlate with the effect of the inhibitor as well, but further investigation is required. The above consideration was added to the discussion (Lines251-255).
As for the reason why only JHOS3 was examined for methylation, the cell line used the most for analysis in this study and easy to handle was chosen as a representative example.
Comment3 (Round1)
SETD8 methylates p53 and this was mentioned in the introduction. p53 methylation should be measured and compared with SETD8 expression/inhibition to tie this into the mechanism of apoptosis regulation. Is p53 methylated in the cell lines tested? Is methylation reduced after SETD8 knockdown and/or inhibition? These mechanistic studies need to be performed for this current manuscript.
Reply3
We appreciate your critical comments and useful suggestions. Although a previous study reported P53 methylation using an antibody, which was designed by Shi, X et al. the antibody is not available for purchase. Therefore, we did not analyze p 53 methylation in this study.
The above consideration was added to the discussion (Lines262-264).